# 3D single-molecule super-resolution microscopy with a tilted light sheet

Anna-Karin Gustavsson[1,2], Petar N. Petrov [1], Maurice Y. Lee[1,3], Yoav Shechtman[1,4] & W.E. Moerner [1,3]

Tilted light sheet microscopy with 3D point spread functions (TILT3D) combines a novel, tilted light sheet illumination strategy with long axial range point spread functions (PSFs) for low-background, 3D super-localization of single molecules as well as 3D super-resolution imaging in thick cells. Because the axial positions of the single emitters are encoded in the shape of each single-molecule image rather than in the position or thickness of the light sheet, the light sheet need not be extremely thin. TILT3D is built upon a standard inverted microscope and has minimal custom parts. The result is simple and flexible 3D super-resolution imaging with tens of nm localization precision throughout thick mammalian cells. We validate TILT3D for 3D super-resolution imaging in mammalian cells by imaging mitochondria and the full nuclear lamina using the double-helix PSF for single-molecule detection and the recently developed tetrapod PSFs for fiducial bead tracking and live axial drift correction.

[1] Department of Chemistry, Stanford University, Stanford, CA 94305, USA. [2] Department of Biosciences and Nutrition, Karolinska Institutet, Stockholm SE-17177 Sweden. [3] Biophysics Program, Stanford University, Stanford, CA 94305, USA. [4]Present address: Biomedical Engineering Department, Technion, Israel Institute of Technology, Haifa 3200003, Israel. Correspondence and requests for materials should be addressed to W.E.M. (email: wmoerner@stanford.edu)

To obtain a complete picture of subcellular structures, cells must be imaged in all three dimensions (3D). Several methods have been developed to extend the imaging capability of single-molecule super-resolution (SR) microscopy[1–3] to 3D. One approach is multiplane imaging[4–7], which requires simultaneous acquisition of multiple images, and was reported to be applicable to an axial range of about 4 μm. A second approach is to use interferometry[8–10], which can result in very high localization precision at the expense of optical complexity and limited axial range per slice. In this work, we use the powerful approach of point spread function (PSF) engineering (for a review, see ref. [11]) that allows for scan-free wide field SR imaging over a several μm axial range per slice. The strategy is to modify the shape of the PSF of the microscope to encode information about the axial ($z$) position of each single emitter directly in its image, which is accomplished by modifying the phase pattern of the emitted light in the Fourier plane of the microscope. This method has been used to create astigmatic PSFs[12,13] and the bisected pupil PSF[14] with axial ranges of 1 μm and 2 μm, respectively, self-bending[15], corkscrew[16], and double-helix (DH) PSFs[17–21] with axial ranges of ~2–3 μm, and the recently developed tetrapod PSFs[22–24], which have a tunable range of up to 20 μm. PSF engineering only requires the addition of a small number of optical elements to the collection path of a standard microscope, making the method relatively simple to implement while exhibiting high precision for 3D single-molecule localization.

A powerful approach to improve imaging in thick cells, specifically the precision of single-molecule localizations, is to reduce the background coming from out-of-focus fluorophores by using light sheet illumination[25,26], where the sample is excited by a thin sheet of light orthogonal to the detection axis. However, early light sheet methods, e.g., selective plane illumination microscopy (SPIM)[25], were designed for low-magnification imaging of large samples, such as embryos. These methods are incompatible with imaging close to a coverslip[27] using a high numerical aperture (NA) imaging objective, which is a requirement for high contrast single-molecule imaging of subcellular structures. Two early methods producing a thin tilted beam are a highly inclined and laminated optical sheet (HILO, pseudo-TIR)[28] and variable-angle epi-fluorescence microscopy (VAEM)[29]. However, in these techniques, the intensity, position, and depth of the pumping light pattern are highly coupled, in contrast to the method presented here. More recently, numerous light sheet designs have been implemented for SR imaging[30–35], but these designs have drawbacks in certain situations. Some designs are incompatible with imaging of fluorophores very close to the coverslip using high NA detection objectives[32,33,36]. In some cases, either the illumination or the detection objective is dipped into the sample chamber[31,32,35,36]. This increases the risk of both biological and fluorophore contaminations of the sample. Some previous designs require complicated optical and electronic apparatus or many custom-made parts, which are often expensive and difficult to build and operate, and thus may not be easily accessible to the general research community.

Here, we present TILT3D, an imaging platform that combines a novel, tilted light sheet illumination strategy with long axial range PSFs. We alleviate many of the aforementioned difficulties in existing light sheet designs by tilting the illumination plane. The tilt allows for sectioning and imaging of cells all the way down to the coverslip. Two perpendicular objectives in close proximity are not required, which enables imaging using a high NA detection objective. No dipping of the objectives into the sample chamber is necessary, which reduces the risk of sample contamination. TILT3D: (a) yields high localization precision of single molecules in 3D over the entire axial range of a mammalian cell via a stack of light sheet slices combined with imaging with engineered PSFs in each slice, (b) has the usual light sheet advantages of reduced photobleaching and photodamage of the sample, and (c) most importantly, is easy and cost-efficient to implement and operate. We validate TILT3D for 3D SR imaging in mammalian cells by imaging mitochondria and the full nuclear lamina using the long-range (2 μm) DH-PSF for SM imaging in each slice and the very long-range (6 and 10 μm) tetrapod PSFs for detection of fiducial beads.

## Results

**TILT3D design and performance**. With TILT3D, there is no need to spend considerable effort and expense producing an extremely thin light sheet, because the 3D positions of the single molecules are not determined from the position of the light sheet but instead from the shape of the DH-PSF. This works with a tilted illumination beam because the DH-PSF is "in focus" with uniform localization precision for a 2 μm axial range even with a high NA collection objective[18]. Therefore, our tilted light sheet was simply created using a cylindrical lens, directed into a glass-walled sample chamber using a prism, and focused at the bottom of the sample chamber using a long working distance illumination objective (Figs. 1 and 2, and Supplementary Fig. 1). The long working distance allows the light sheet to be introduced without dipping the objective into the sample chamber, reducing the risk of any biological or fluorescent sample contamination. The sample chamber can be left open to allow for easy sample access, or sealed to prevent sample contamination and evaporation of medium, and to limit access of oxygen to the sample. A 10° downward tilt of the light sheet enables the light sheet to be introduced into the sample chamber far away from the distorting bottom glass–water interface, while allowing illumination even at the bottom surfaces of adherent cells. The resulting light sheet has a thickness (waist radius) of 2.1 μm and width of 19 μm ($1/e^2$), and a confocal parameter of 73 μm (Supplementary Fig. 2a, b), making it well suited for studies of mammalian cells. The light sheet is scanned stepwise in 1 μm steps in the axial direction to acquire slices which section the cell using a motorized tiltable mirror positioned in the conjugate plane to the back aperture of the illumination objective (Fig. 2 and Supplementary Figs. 2c and 3). The two illumination modes compared here (wide field epi-illumination and light sheet) were easily toggled with a flip mirror.

A lens pair in 4$f$ configuration was added to the emission side of the microscope to implement PSF engineering, i.e., phase modulation of the light in the Fourier plane of the microscope. Phase modulation was accomplished using transmissive dielectric phase masks or a deformable mirror (DM) to create the 3D PSFs (Fig. 1 and Supplementary Fig. 4). The long axial range of the DH-PSF works well with a 2-μm thick light sheet to observe bright molecules in the central 1 μm region over a ~15 × 15 μm transverse region because the dim molecules in the edges of the light sheet in $z$ are not bright and are not used. The DH-PSF consists of two lobes instead of just one, where the midpoint between the two lobes reports on the $xy$ position and the $z$-position is determined from the angular orientation of the line connecting the center of the lobes. The very long-range tetrapod PSFs describe a more complex pattern, roughly tracing out the shape of a tetrapod when envisioned in 3D. A key advantage of the tetrapod PSFs are their smoothly varying phase patterns that can be generated with a DM, as demonstrated here for the first time, rather than using a microfabricated dielectric mask or an inherently photon inefficient liquid crystal spatial light modulator. The resulting PSFs are both photon efficient and easily tunable for different depth ranges. Even though the very long-range tetrapod has a larger transverse footprint, it is quite useful

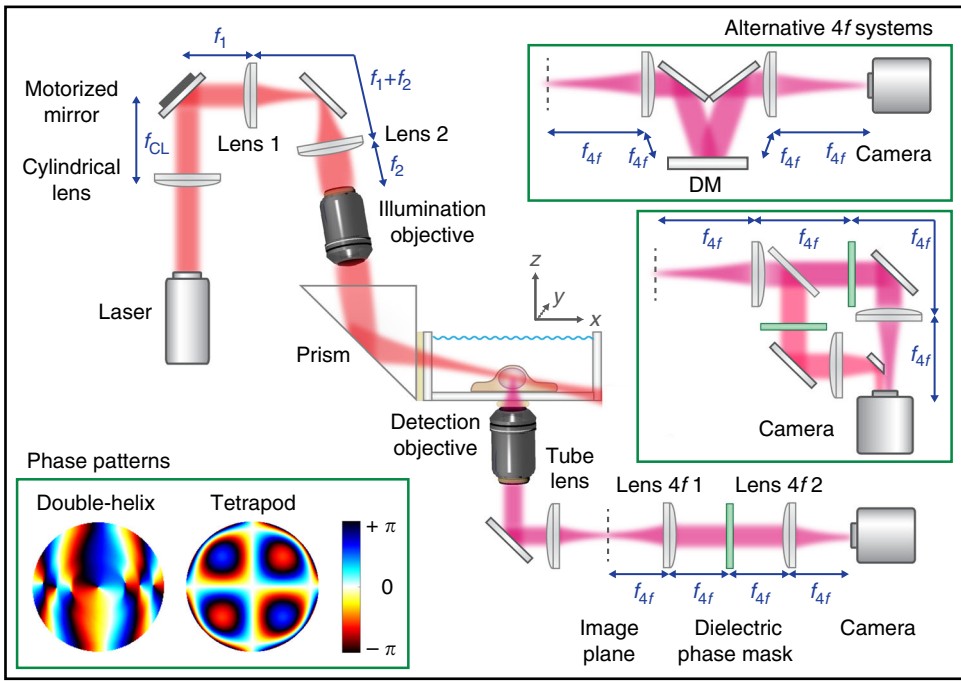

**Fig. 1** Simplified schematic of the TILT3D design. The light sheet is formed by a cylindrical lens, relayed to the back aperture of a long working distance illumination objective, and reflected at an angle into the sample using a glass prism on the outer side of the imaging chamber. A motorized mirror is used to adjust the $y$- and $z$-position of the light sheet. The emitted light is imaged through a $4f$ system, where a transmissive dielectric phase mask or a deformable mirror (DM) is placed in the Fourier plane for phase modulation. The phase pattern reshapes the point spread function (PSF) to encode the axial position of the emitter. Lower left inset shows the phase patterns for a double-helix PSF, which provides in-focus images over 2 μm axially, and a tetrapod PSF with 6 μm axial range. Upper right insets show alternative $4f$ systems when using a DM in the Fourier plane and when using two channels with transmissive phase masks

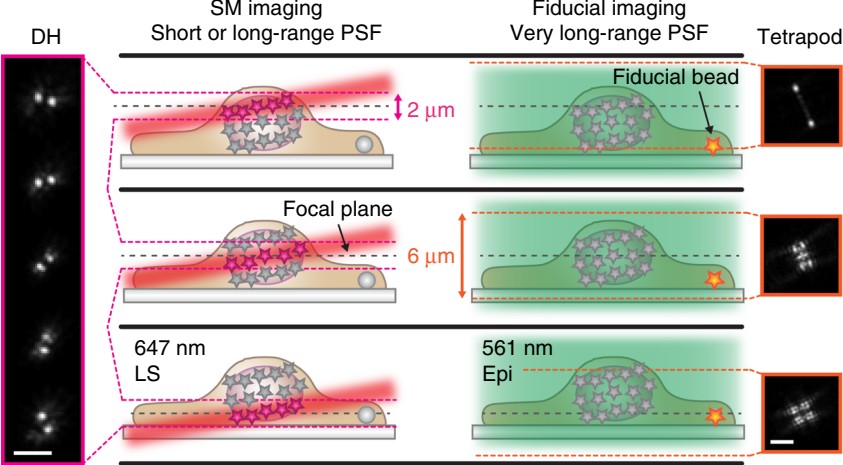

**Fig. 2** Schematic of interleaved illumination and detection scheme. Single molecules (SMs) were excited using light sheet (LS) illumination and detected using either a conventional clear aperture point spread function (PSF) or the double-helix (DH) PSF. SM imaging was interleaved with imaging of a fiducial bead that could be positioned anywhere in the field of view. The fiducial bead was excited using epi-illumination (Epi) and detected using either the DH-PSF or a very long axial range tetrapod PSF to allow illumination and detection independent of the axial position of the bead. The fiducial bead was localized in real time and the sample drift was corrected in the axial direction. The illumination was also interleaved with SM reactivation using a 405 nm laser when needed. In the example shown here, SMs over 2-μm axially were detected with the DH-PSF and fiducial beads imaged with a 6-μm axial range tetrapod PSF. The schematic is not to scale. Scale bars are 3 μm

for imaging very sparse, bright objects[23], thus we use this PSF for imaging of fiducial beads in a thick cell sample (*vide infra*). A comparison between experimental and modeled PSFs for the case of a 6-μm axial range tetrapod is shown in Supplementary Fig. 5.

**2D imaging with live axial drift correction.** The setup performance was benchmarked in terms of contrast improvement by 2D bulk and single-molecule imaging of individual light-sheet-defined slices of lamin B1 in HeLa cells immunolabeled with Alexa Fluor 647. First, we used a DM to generate a clear aperture

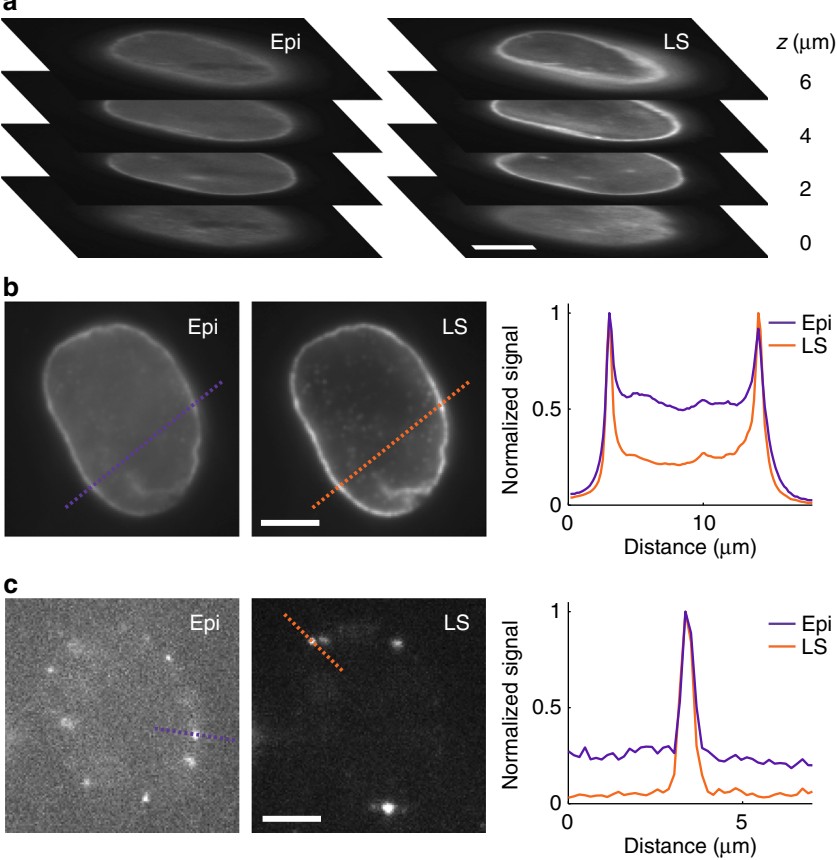

**Fig. 3** Sectioning using light sheet illumination improves contrast in 2D imaging. **a** Comparison between epi- (Epi) and light sheet (LS) illumination at different *z*-positions throughout a cell nucleus. The image plane was moved using the piezoelectric objective scanner, and for LS imaging the LS plane was moved together with the image plane using the motorized mirror. The sectioning capability of the LS is clearly demonstrated, showing an improved contrast when compared with epi-illumination for all cell sections. **b** Diffraction-limited and **c** single-molecule images demonstrating that light sheet illumination improves the signal-to-background ratio up to fivefold compared to conventional epi-illumination. Graphs show line scans over the dashed lines in the images. All images show lamin B1 immunolabeled with Alexa Fluor 647 in HeLa cells. Compared images are shown with the same linear grayscale, respectively. Scale bars are 5 μm

PSF for conventional single-molecule detection in the middle of the nucleus. Figure 3a, b shows a clear contrast improvement for all slices even for diffraction-limited imaging. Turning to single molecules, Fig. 3c shows that light sheet illumination yields up to a fivefold increase in the signal-to-background ratio (SBR) relative to conventional epi-illumination, resulting in improved localization precision for 2D (or 3D) imaging. We note that by adding fluorophore blinking (Supplementary Movie 1), these 2D images can yield super-resolution information. Over the time required to acquire the large number of single-molecule images needed for a super-resolution reconstruction, the sample can drift several hundreds of nm in all three dimensions. Such drift would severely degrade the reconstruction and blur nanoscale features. To facilitate 3D drift correction, we use a 10-μm axial range tetrapod PSF implemented on the DM for interleaved detection of a fiducial bead at the coverslip. The flexible scheme outlined in Fig. 2 makes it easy to alternate the imaging of the nuclear lamina and the fiducial bead several μm below at the coverslip surface while keeping the focal plane stationary at the middle of the cell. This reduces the risk of microscope drift, and it also simplifies the analysis of the absolute position of the fiducial bead. A super-resolution 2D image of a lamina cross-section is shown in Supplementary Fig. 6, where line scanning over the structure yielded a measured lamina width of 130 nm (FWHM). The analysis methodology used for fiducial bead detection with the tetrapod PSF is outlined in Supplementary Fig. 7.

**3D imaging with long axial range DH-PSF**. Next, the DM was replaced by a transmissive DH dielectric phase mask for single-slice 3D SR imaging of the mitochondrial outer membrane in HeLa cells, visualized by TOM20 immunolabeled with Alexa Fluor 647 (Fig. 4a, Supplementary Fig. 8, and Supplementary Movies 2–5). In this case, far red fiducial beads were used, allowing for continuous excitation using 647 nm and detection using the DH-PSF. Cross-sections reveal the hollow structure of the mitochondrial outer membrane (Fig. 4b). More examples of individual reconstructed mitochondria are shown in Supplementary Fig. 8. This imaging scheme was very easy to implement and required no scripts to control the setup and only a single laser for illumination.

Mitochondria in HeLa cells were also imaged with the DH-PSF when switching from epi-illumination to light sheet illumination to allow for direct comparison between the two illumination modes. This resulted in localization precisions when using epi/light sheet illumination of 23/16 nm in *xy* and 35/24 nm in *z*, respectively, demonstrating considerable improvement when using light sheet illumination (Supplementary Fig. 9 and Supplementary Movie 6).

Finally, 3D SR imaging of the entire nuclear lamina in HeLa cells immunolabeled with Alexa Fluor 647 was performed using the interleaved illumination scheme shown in Fig. 2 with live axial drift correction (Fig. 4c, Supplementary Fig. 10, and Supplementary Movies 7–9). In this case, the 4*f* system was

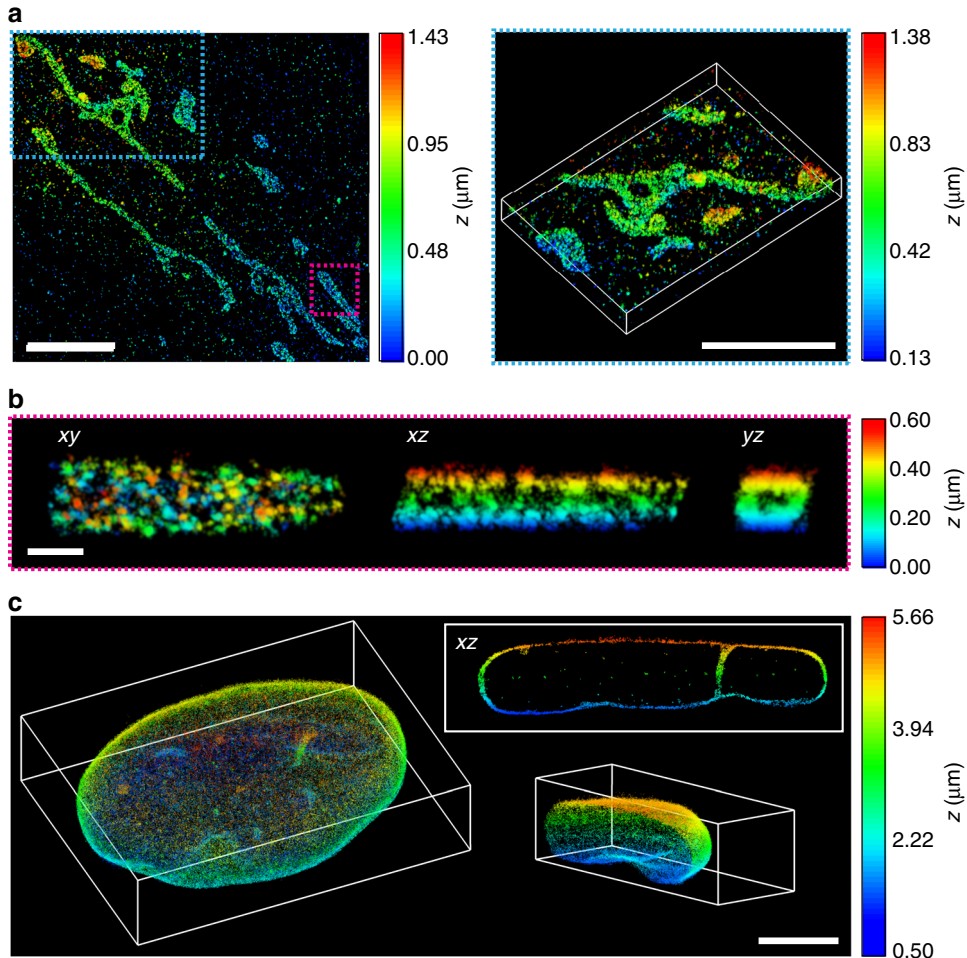

**Fig. 4** Tilted light sheet imaging with long axial range point spread functions (PSFs). **a** 3D super-resolution (SR) reconstructions of mitochondria (TOM20) in a HeLa cell. The double-helix (DH) PSF was used for imaging of both single molecules and fiducial beads. **b** *xy*, *xz*, and *yz* views of the mitochondrion shown in the magenta rectangle in **a**, revealing the hollow cylinder structure of the mitochondrial outer membrane. **c** 3D SR reconstruction of the entire nuclear lamina (lamin B1) in a HeLa cell. Imaging of single molecules and fiducial beads was performed with the DH-PSF and a 6-μm tetrapod PSF, respectively. The *xz* view shows a 1.3-μm thick *y*-slice through the cell, where lamin meshwork enveloping an intranuclear channel is visible. The lower right inset shows the right cap of the reconstruction. All samples imaged were immunolabeled with Alexa Fluor 647. Scale bars are 5 μm in **a** and **c**, and 500 nm in **b**

extended to two channels, where single molecules were detected using the long-range DH-PSF in the far red channel and several fiducial beads were detected in the red channel using the very long-range 6-μm tetrapod PSF implemented using transmissive phase masks. The focal plane was moved in 1 μm steps together with the light sheet to sample the entire nuclear lamina in several thick, overlapping slices. The very long axial range of the tetrapod PSF allowed the fiducial beads to be detected and localized in all slices. At 3.3 μm above the coverslip, the localization precision was estimated to 16 nm in xy and 23 nm in *z* for SM detection using the DH-PSF, and 3 nm in *xy* and 7 nm in *z* for fiducial bead detection using the 6-μm tetrapod PSF (Supplementary Fig. 11a, b). The thickness of the lamina at the bottom and the top of the cell was measured to be 113 nm and 101 nm (FWHM), respectively (Supplementary Fig. 11c). An *xz* view reveals the lamin meshwork enveloping an intranuclear lamin channel (Fig. 4c and Supplementary Movie 10).

## Discussion

By combining a tilted light sheet with PSF engineering, TILT3D offers a simple yet powerful tool for 3D SR imaging in whole

mammalian cells using only a few axial light sheet positions. No scanning of the detection objective is required over the entire range of the PSF. The flexible design allows for imaging close to a conventional coverslip using a high NA detection objective and easy switching between illumination modes, lasers, and PSFs. The implementation of the simple tilted light sheet in combination with PSF engineering drastically improves the localization precision of single molecules as compared to when using conventional epi-illumination.

3D PSFs cover a larger transverse spatial extent than the standard PSF, which can reduce SBR, but this issue is not too critical for the DH-PSF used for single-molecule imaging. Of course, using light sheet illumination for single-molecule imaging already reduces out-of-focus background. The spatial extent of a 3D PSF can also require the density of emitters to be sparser than for 2D imaging with the standard PSF. For the DH-PSF with 2 μm range, this can be easily achieved by initially forcing a larger fraction of the fluorophores into a dark state. Sealing of the cell chamber in combination with light sheet illumination ensures that the bleaching of fluorophores is kept to a minimum even during extended acquisition times. When using the live drift correction scheme presented here, the image acquisition requires minimal user input or presence.

The 6-μm and 10-μm tetrapod PSFs were shown to be excellent for fiducial bead localization, yielding localization precisions of a few nanometers and which can be acquired for any position of the fiducial beads in a mammalian cell sample. This is quite useful, since fiducial beads often do not coincide with the desired image plane and can be situated either at the coverslip, on top of the cells, or anywhere inside the cells due to endocytosis. Having a long axial range for fiducial detection also allows for tracking of the same fiducials over several different image slices, thus providing one or several fixed points throughout the entire reconstruction. Although drift correction can be implemented in a number of different ways[7], the strategy outlined here provides a simple and robust method, which is easy to implement on any microscope.

The platform can be extended to multi-color light sheet imaging and to more advanced light sheet techniques, such as a scanned Gaussian, Bessel[37], or Airy beams[38], by replacing the motorized mirror with an acousto-optical deflector or a spatial light modulator. We demonstrate here that 3D SR imaging successfully can be performed in thick samples, such as the nuclear lamina, using an oil immersion imaging objective that is often used. The intrinsic flexibility of the TILT3D design allows the imaging objective to be easily replaced with a water or silicone oil immersion objective in cases where aberrations caused by index mismatch in the sample must be reduced. Furthermore, the DM can be used for adaptive optics (AO) to correct system aberrations if needed[39], but it is important to remember that the DH-PSF phase pattern itself contributes the dominant aberration so that a simple double Gaussian estimator can be used effectively throughout the sample, achieving comparable performance at the top and bottom of the nucleus. We believe that TILT3D in the future will become an important tool not only for 3D SR imaging, but also for live whole-cell single-particle and single-molecule tracking.

## Methods

Additional methods and any associated references are available in the Supplementary Information.

**Optical setup.** In this work, the optical setup was built around a conventional inverted microscope (IX71, Olympus) (Fig. 1 and Supplementary Fig. 1). Illumination lasers (405 nm, 100 mW; 561 nm, 200 mW; and 647 nm, 120 mW, all CW, from Coherent) were spectrally filtered (561 nm: ff01-554/23-25 excitation filter, 647 nm: ff01-631/36-25 excitation filter, both Semrock), circularly polarized (LPVISB050-MP2 polarizers, Thorlabs, and 405 nm: Z-10-A-.250-B-405 quarter-wave plate, Tower Optical, 561 nm: WPQ05M-561 quarter-wave plate, Thorlabs, 647 nm: WPQ05M-633 quarter-wave plate, Thorlabs), and expanded and collimated using lens telescopes. The toggling of the lasers was controlled with shutters (VS14S2T1 with VMM-D3 three-channel driver, Vincent Associates Uniblitz) and synchronized with the detection optics via MATLAB. The 405 and 561 nm lasers were introduced into the back port of the microscope through a Köhler lens to allow for wide field epi-illumination. The 647 nm laser was either introduced into the epi-illumination pathway or sent to the light sheet illumination pathway; the pathway was easily switched with a flip mirror. The light sheet illumination pathway consisted of a cylindrical lens (LJ1558L2-A, f = 300 mm, Thorlabs, or ACY254-200-A, f = 200 mm, Thorlabs), which focused the light in only one dimension onto a motorized mirror (8821 mirror mount with 8742 Picomotor controller, Newport). The motorized mirror plane was imaged onto the back aperture of a long working distance illumination objective (378-803-3, x10, NA 0.28, Mitutoyo) by two lenses in a 4f configuration. The illumination objective then focused the light sheet, which was directed into the sample chamber (704-00-20-10, Hellma) at an angle of about 10° using a glass prism (PS908L-A, Thorlabs). The entire light sheet illumination path was mounted on a breadboard above the microscope stage, which could be moved by an xyz translation stage (460P, Newport).

The emitted light from the fluorophores was detected by a high NA detection objective (UPLSAPO100XO, x100, NA 1.4, Olympus) mounted on a piezoelectric objective scanner (Nano-F100, C-Focus, Mad City Labs), spectrally filtered (Di01-R405/488/561/635 dichroic, for far red detection: Semrock, ET700/75m bandpass filter, Chroma, ZET647NF notch filter, Chroma, 3RD650LP longpass filter, Omega Optical, and for red detection: ZET647NF notch filter, Chroma, et610/60 bandpass filter, Chroma, and FF01-593/40 bandpass filter, Semrock), and focused by the

microscope tube lens, before entering a 4f imaging system. The first lens of the 4f system (f = 150 mm when using a deformable mirror and f = 90 mm when using transmissive phase masks) was positioned one focal length from the intermediate image plane formed by the microscope tube lens. In the plane one focal length after the first 4f lens (i.e., the Fourier plane of the imaging path), the phase of the emitted light was modulated to reshape the PSF to encode the axial position of the emitter. After phase modulation, the light was focused by the second 4f lens and imaged using an EM-CCD camera (iXon3 897, Andor). To create a two-channel 4f system (Fig. 1, lower right inset) a dichroic mirror (T660lpxrxt, Chroma) was inserted before the phase masks to transmit far red light into the first light path and reflect light with wavelengths shorter than 660 nm into a second light path. The paths were merged before the camera using a D-shaped mirror (BBD1-E02, Thorlabs). The desired phase pattern was implemented either using transmissive dielectric phase masks (for the DH mask, Double-Helix Optics, LLC, and the Tetrapod phase mask was fabricated as outlined in the Supplementary Information), or using a deformable mirror (DM) (Multi-DM 3.5, Boston Micromachines Corporation). Transmissive phase masks were used to implement the spatially non-smooth phase pattern of the DH-PSF. Since the DM consists of a continuous membrane, it is more suited to encoding smoothly varying phase patterns, such as those of the tetrapod PSFs. When the transmissive phase masks were used, the physical mask had to be exchanged whenever there was an adjustment to be made to the desired working axial range of the PSF and/or the wavelength of the emitted light. To facilitate these exchanges, the transmissive phase masks were mounted on magnetic mounts. When the DM was used, the phase pattern was controlled from a computer and could be rapidly updated with different phase patterns to produce PSFs with different axial ranges and/or at different wavelengths of light. Another common way to implement the phase pattern is to use a liquid crystal spatial light modulator (SLM). However, replacing the SLM with a transmissive phase mask or a DM substantially increases photon efficiency, which is a limiting factor for obtaining better precision for single-molecule imaging. The loss of the unmodulated polarization with a liquid crystal SLM can, however, be recovered[40]. The active regions of the transmissive masks and the DM had diameters of 2.7 and 4.2 mm, respectively. To ensure that the electric field diameter at the Fourier plane, $d_E$, matched the diameter of the active regions, the focal lengths of the 4f lenses, $f_{4f}$, had to be chosen accordingly. Under the Abbe sine condition, the electric field diameter depends on the focal length according to

$$d_E = \frac{2f_{4f}\mathrm{NA}}{\sqrt{M^2 - \mathrm{NA}^2}} \qquad (,1)$$

where NA = 1.4 is the numerical aperture of the detection objective and M = 100 is the magnification of the microscope. Choosing focal lengths of the 4f lenses of 90 and 150 mm for the transmissive masks and DM, respectively, resulted in electric field diameters of 2.5 and 4.2 mm, which matched or were slightly smaller than the active mask regions. Overfilling the mask would lead to unmodulated light in the image plane.

**Prism and sample chamber assembly.** The glass prism for reflection of the light sheet into the chamber was attached to a standard microscope stage using a custom-made right triangular aluminum prism for support. This is the only custom-made component of the entire imaging platform (other than the tetrapod phase mask). The aluminum support was cut from a square aluminum bar and designed such that when the glass prism was attached to it, the back surface of the glass prism was exposed to air (instead of aluminum) to facilitate total internal reflection at the glass–air interface (Supplementary Fig. 1b). The aluminum prism was attached to the stage and the glass prism to the aluminum support using two-part silicone rubber (Ecoflex® 5, Reynolds Advanced Materials). The sample chamber was assembled by attaching a glass coverslip (Fisher Premium Cover Glass, no. 1.5) with cells cultured on it to the bottom of the four transparent polished walls of a sliced commercial glass cuvette (704-000-20-10, Hellma) using two-part silicone rubber. Using a standard glass coverslip allowed for easy cell culturing and handling, and facilitated the usage of a high NA detection objective for SR imaging. During imaging, the sample chamber could be left open on the top to allow for easy sample access, or it could be easily sealed by placing a second coverslip on top of the four walls and sealing it with silicone rubber. This reduced the risk of sample contamination, decreased the rate of evaporation of the medium, and limited the access of oxygen to the sample. The glass surfaces between the coverslip and the imaging objective, and between the sample chamber and the glass prism, were brought into optical contact with immersion oil (Zeiss Immersol 518F, n = 1.518). Clearly, index matching gel could also be used between the sample chamber and the glass prism. After imaging, the bottom coverslip could be removed and the chamber walls cleaned and reused.

**Cell culture and seeding.** HeLa cells were cultured at 37 °C and 5% $CO_2$ in high-glucose Dulbecco's modified Eagle's medium (DMEM, HyClone) supplemented with 10% (v/v) FBS (HyClone). Two days before imaging, cultured cells were plated onto plasma-etched coverslips (Fisher Premium Cover Glass, no. 1.5) spun coat with a 1% (w/v) polyvinyl alcohol (PVA, Polysciences Inc.) layer containing red (lamin B1) (580/605 nm, F8810, Invitrogen) or far red (mitochondria) (625/645 nm, F8806, Invitrogen) fluorescent microspheres, cultured for 24 h in high-glucose

DMEM supplemented with 10% FBS, and subsequently cultured for 24 h in high-glucose, phenol-red-free DMEM (HyClone) supplemented with 10% FBS. During this period, some of the microspheres were endocytosed.

**Cell fixation and immunolabeling**. Immunolabeling steps were performed with coverslips placed on PARAFILM, while all other steps were performed with coverslips placed in 6-well plates. After the immunolabeling, all samples were protected from light and stored in PBS (HyClone) at 4 °C. The samples were imaged within 48 h of labeling.

For lamin B1 labeling, HeLa cells (CCL-2, ATCC) were washed three times in PBS, fixed in chilled 4% paraformaldehyde (PFA) (Electron Microscopy Sciences) in PBS for 20 min, washed once in PBS, and incubated with 10 mM NH$_4$Cl (Sigma-Aldrich) in PBS for 10 min. Next, the cells were permeabilized with three washing steps with 0.2% (v/v) Triton X-100 (Sigma-Aldrich) in PBS with 5 min incubation between each wash, and blocked with 3% (w/v) BSA (Sigma-Aldrich) in PBS for 1 h. The cells were then labeled with primary rabbit anti-lamin B1 (ab16048, Abcam) using a 1:1000 dilution in 1% (w/v) BSA in PBS for 2 h, washed three times with 0.1% (v/v) Triton X-100 in PBS with 3 min incubation between each wash, and labeled with secondary donkey anti-rabbit conjugated with Alexa Fluor 647 (ab150067, Abcam) using a 1:1000 dilution in 1% (w/v) BSA in PBS for 1 h. The cells were finally washed five times with 0.1% (v/v) Triton X-100 in PBS with 1 min incubation between each wash.

For immunolabeling of mitochondria, HeLa cells (CCL-2, ATCC) were washed two times in PBS, fixed in chilled 4% PFA in PBS for 20 min, washed once in PBS, and incubated with 10 mM NH$_4$Cl in PBS for 10 min. Next, the cells were permeabilized with three washing steps with 0.2% (v/v) Triton X-100 in PBS with 5 min incubation between each wash, and blocked with 3% (w/v) BSA in 0.1% (v/v) Triton X-100 in PBS for 1 h. The cells were labeled with Alexa Fluor 647-conjugated primary rabbit anti-TOMM20 (ab209606, Abcam) using a 1:100 dilution in 1% (w/v) BSA in 0.1% (v/v) Triton X-100 in PBS for 2 h. The cells were then washed five times with 0.1% (v/v) Triton X-100 in PBS with 3 min incubation between each wash, post-fixed in 4% PFA in PBS for 5 min, and washed three times in PBS.

**Single-molecule super-resolution imaging**. For diffraction-limited imaging, cells were imaged in PBS using low power 647 nm excitation (~1 W cm$^{-2}$). Comparisons between light sheet and epi-illumination were performed by manually switching illumination light paths using a flip mirror during image acquisition. Custom scripts were written in MATLAB to synchronize the laser shutters, phase patterns on the DM, image acquisition on the camera, and translation by the piezoelectric objective scanner. Scanning of the light sheet was performed using the New Focus Picomotor Application software (Newport).

For single-molecule SR imaging, the PBS was replaced by a reducing and oxygen-scavenging buffer[41] comprising 100 mM Tris-HCl (Invitrogen), 10% (w/v) glucose (BD Difco), 2 µl ml$^{-1}$ catalase (Sigma-Aldrich), 560 µg ml$^{-1}$ glucose oxidase (Sigma-Aldrich), and cysteamine (Sigma-Aldrich) with a concentration of 10 mM (Alexa Fluor 647 immunolabeled lamin B1 (2D)), 20 mM (Alexa Fluor 647 immunolabeled mitochondria), or 40 mM (Alexa Fluor 647 immunolabeled lamin B1 (3D)). At the beginning of the measurement, a large fraction of the Alexa Fluor 647 molecules were converted into a dark state using 647 nm epi-illumination at 5 kW cm$^{-2}$. For all single-molecule measurements, an exposure time of 50 ms and a calibrated EM gain of 186 was used. For each imaging experiment, at least 300 dark frames were acquired when the shutter of the camera was closed. The mean of these dark frames was subtracted from the images before analysis. When using 3D PSFs, calibration of each PSF was carried out by axial scanning of the fiducial beads in the sample over the full PSF range using the piezoelectric objective scanner.

For acquisition of 2D SR images of lamin B1 in HeLa cells, a 4f system with a DM was used for phase modulation in the imaging pathway. A standard PSF was used together with ~25 kW cm$^{-2}$ 647 nm light sheet illumination for imaging of single Alexa Fluor 647 molecules, while for every twentieth frame a fiducial bead at the coverslip was imaged using a 10-µm tetrapod PSF and 10 W cm$^{-2}$ 561 nm epi-illumination (Fig. 2). This scheme makes it easy to alternate the imaging of the nuclear lamina and of the fiducial bead several microns below at the surface of the coverslip while keeping the focal plane stationary. Using a fluorecent bead that is excited by a different wavelength reduced the risk of photobleaching the fluorophores and of saturating the single-molecule image by light from the fiducial bead, while maintaining a high photon count for the fiducial. The fiducial frame was analyzed in real time and the sample drift corrected in the axial direction via feedback to the piezoelectric objective scanner. Drift correction in the image plane was performed during post processing, but could, in principle, have been performed live during imaging via feedback to a motorized stage. Reactivation of the Alexa Fluor 647 molecules back from the dark state was performed with 3 W cm$^{-2}$ 405 nm epi-illumination every hundredth frame after the first 10,000 frames were acquired. Single-molecule data comparing epi-illumination and light sheet illumination is shown in Supplementary Movie 1.

For 3D SR imaging of mitochondria in HeLa cells, the DM was replaced by a transmissive dielectric phase mask (Supplementary Movie 2). In this case, far red fiducial beads were used, allowing for continuous excitation using 647 nm and detection using the DH-PSF. Since the range of the PSF covered the entire range of the sample, no live drift correction was necessary. This scheme was very easy to

implement and required no scripts to control the setup and only a single laser for illumination.

Mitochondria in HeLa cells were imaged with the DH-PSF while switching from epi-illumination to light sheet illumination to allow for direct comparison between the signal, background, and localization precision in the two cases (Supplementary Movie 6).

Acquisition of 3D SR images of lamin B1 in HeLa cells was performed using the scheme shown in Fig. 2 using interleaved illumination and live axial drift correction. In this case, the 4f system was extended to two channels, where single molecules were detected using the DH-PSF in the far red channel and the fiducial beads were detected in the red channel using a 6-µm tetrapod PSF, both implemented using transmissive phase masks. Several slices were aqcuired and the focal plane was moved together with the light sheet to sample the entire nuclear lamina. The positions of the fiducial beads, which were detectable in all slices owing to the very long axial range of the tetrapod PSF, were used during post processing to correct lateral and axial drift within each slice and to stitch together the different slices. The final z-position of each slice was corrected using cross-correlation between two adjacent slices.

**Code availability**. The custom-written code generated during the current study is available from the corresponding author on request. Calibration and fitting of tetrapod fiducial images for drift correction was performed using a modified version of the open-source Easy-Pupil-Finder software[42] (https://sourceforge.net/projects/easy-pupil-finder/). Calibration and fitting analysis of DH-PSF images was performed using a modified version of the open-source Easy-DHPSF software[43] (https://sourceforge.net/projects/easy-dhpsf/).

**Data availability**. The single-molecule localizations generated and analyzed during the current study are available from the corresponding author on request.

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

## Acknowledgements

This work was supported in part by the National Institute of General Medical Sciences (Grant No. R35GM118067) to W.E.M. and by the National Institute of Biomedical Imaging and Bioengineering (Grant No. U01EB021237) to W.E.M. A.-K.G. acknowledges partial financial support from the Swedish Research Council (Grant No. 2016-00130), and from the Foundation BLANCEFLOR Boncompagni-Ludovisi, née Bildt. M. Y.L. is supported by a National Science Scholarship (PhD) from A*STAR, Singapore. Y.S. is supported in part by a Career Advancement Chairship from the Technion. We also thank Dr Carl G. Ebeling, Worldwide Application Scientist for Bruker Fluorescence Microscopy, for his support and for the use of the Bruker SRX visualization and analysis software for rendering localization data, and Dr Steffen Sahl for early discussions. Fabrication of dielectric phase masks was performed at the Stanford Nanofabrication Facility, which is supported by the National Science Foundation as part of the National Nanotechnology Coordinated Infrastructure under award ECCS-1542152.

## Author contributions

A.-K.G., P.N.P., and M.Y.L. developed and constructed the imaging platform. A.-K.G. labeled the cells, planned and performed the experiments, and analyzed the 2D and double-helix data. P.N.P. and M.Y.L. helped performing the experiments. P.N.P. and Y.S. developed the tetrapod analysis algorithms and P.N.P. analyzed the tetrapod data. M.Y.L. fabricated the transmissive tetrapod phase mask. Y.S. developed the tetrapod phase patterns and assisted with their implementation on the deformable mirror. W.E.M. conceived the idea and supervised the research. All authors contributed to writing the paper.

## Additional information

**Competing interests:** The authors declare no competing financial interests.

