## [Peer Review File · Nature Communications]

Reviewers' comments:

Reviewer #1 (Remarks to the Author):

It is a very good paper having an important impact on the super resolved microscopy community. However, I would suggest adding a demonstration of the validity and robustness of the approach by performing 3D imaging in the cell nucleus of the distribution of the histone H2B tagged by GFP. In this case, the scenario changes with respect to the distribution of the fluorescent molecules in a crowded environment and background could play a different role. In any case, it is a paper that deserves to be published in this journal.

Reviewer #2 (Remarks to the Author):

3D single-molecule super-resolution microscopy with a tilted light sheet

The manuscript by Gustavsson et al. reports the combination of tilted light sheet microscopy and PSF engineering to achieve super-resolution images by single-molecule detection. Two different phase masks were used to detect single particles: 1) track fiducial markers over long distances in Z (across 6 μ m) and 2) detect single molecules in mammalian cells (across 2 μ m). They further utilized a tilted light sheet to illuminate regions close to the coverslip surface and increase the SBR. The authors argue that the combination with PSF engineering does not require a thin light sheets which also eases the hardware implementation for fellow researchers.

The manuscript is nicely written and very detailed in the presented work. While most of the technical aspects and biological applications have been reported elsewhere, I find the idea appealing to use different phase masks in order to gain accurate positional information from various distances.

However, the authors should discuss the advantage of using the two different phase masks over the technique used by, for example, Hajj et al. (Ref. 7). They illuminated a polystyrene bead with infrared light to use it as the fiducial marker.

It would be very interesting for the reader to see the advantage/difference in resolution (drift) with and without fiducial marker in text or figure.

Also, did the authors consider highly inclined and laminated optical sheet microscopy (HILO) for the illumination? The method is readily installed for inverted microscopes, does not need any additionally optical parts, and gives superior SBR. If the method is not suitable, the authors should maybe discuss in the text.

For the lack of novelty, I cannot recommend the current manuscript for publication in Nature Communications.

Minor points:

1 In the section "3D imaging with long axial range DH-PSF", it is not clear to me why the authors chose to exchange the DM by the DH phase mask. The section should be written more clearly.

2 Ref. 35 is not properly formatted

3 A similar publication describes the use of light-sheet microscope and PSF engineering and should be considered for citation (Bin Yu, Yu, J., Li, W., Cao, B., Li, H., Chen, D., & Niu, H. (2016). Nanoscale three-dimensional single particle tracking by light-sheet-based double-helix point spread function microscopy. *Applied Optics*, 55(3), 449–453)

Reviewer #3 (Remarks to the Author):

The authors address an important problem in their manuscript the imaging of single molecules in three dimensions for the purpose of super resolution microscopy. In 3D samples background fluorescence can be a significant problem. Therefore selective illumination approaches have been proposed over recent years to reduce such out of focus background signals. Light sheet microscopy has received considerable attention with several different approaches having been published. The authors add a new approach which combines light sheet microscopy on the excitation side with the detection of engineered point spread functions.

The paper is very carefully written and every step is well explained. Considering the number of alternative approaches using light sheet illumination that have already been published, I would, however, have expected a more detailed discussion of the pros and cons of this new approach to implementing light sheet microscopy.

Reviewer 1

We thank the reviewer for the time spent on our manuscript as well as the highly positive response.

Below, we provide responses (in red) to each of the questions/comments (*in black italic*), with our additions to the text indicated (in blue).

- 1. It is a very good paper having an important impact on the super resolved microscopy community. However, I would suggest adding a demonstration of the validity and robustness of the approach by performing 3D imaging in the cell nucleus of the distribution of the histone H2B tagged by GFP. In this case, the scenario changes with respect to the distribution of the fluorescent molecules in a crowded environment and background could play a different role. In any case, it is a paper that deserves to be published in this journal.*

It is a nice suggestion to image a sample where all H2Bs are labeled with GFP. This kind of sample would indeed provide imaging conditions with high background and fluorophore density. It is worth remembering that the sectioning capability of the light sheet would improve the signal-to-background ratio as compared to epi-illumination also in this case. The increase in fluorophore density would require a larger fraction of the fluorophores to initially be shelved into a dark state for successful imaging of sparse single molecules. A further issue is the relatively poor blinking of GFP, but this could be addressed by a photoactivatable GFP mutant or by using a nanobody to GFP with a small-molecule blinker such as AlexaFluor-647 (as used in this paper). Basically, for an experiment on such an additional sample, the principles would be identical to those demonstrated here. We therefore consider an extra imaging demonstration of this additional sample to be somewhat redundant.

Reviewer 2

We thank the reviewer for examining the manuscript in detail and for providing useful comments. We have addressed each comment to improve the paper in all aspects, yielding a much clearer demonstration of the benefits of TILT3D.

Below, we provide responses (in red) to each of the questions/comments (*in black italic*), with our additions to the text indicated (in blue).

2. *3D single-molecule super-resolution microscopy with a tilted light sheet*

The manuscript by Gustavsson et al. reports the combination of tilted light sheet microscopy and PSF engineering to achieve super-resolution images by single-molecule detection. Two different phase masks were used to detect single particles: 1) track fiducial markers over long distances in Z (across 6um) and 2) detect single molecules in mammalian cells (across 2um). They further utilized a tilted light sheet to illuminate regions close to the coverslip surface and increase the SBR. The authors argue that the combination with PSF engineering does not require a thin light sheets which also eases the hardware implementation for fellow researchers.

The manuscript is nicely written and very detailed in the presented work. While most of the technical aspects and biological applications have been reported elsewhere, I find the idea appealing to use different phase masks in order to gain accurate positional information from various distances.

However, the authors should discuss the advantage of using the two different phase masks over the technique used by, for example, Hajj et al. (Ref. 7). They illuminated a polystyrene bead with infrared light to use it as the fiducial marker.

In terms of comparing the two methods for fiducial localization, we and Hajj have a similar approach. Specifically, to get precise axial localization for the bead, Hajj used the rings in the out-of-focus IR image to maintain focus (see Fig S5 of Hajj). This approach is not too different from using a highly structured tetrapod PSF like we use. Therefore we are happy to cite this paper again in the Discussion section of the manuscript to clarify this point:

Although drift correction can be implemented in a number of different ways [Hajj et al. 2014], the strategy outlined here provides a simple and robust method which is easy to implement on any microscope.

3. *It would be very interesting for the reader to see the advantage/difference in resolution (drift) with and without fiducial marker in text or figure.*

It is well-known that drift correction is often critical in super-resolution imaging, so we are happy to clarify this by adding the following sentence to the *2D imaging with live axial drift correction* section of the manuscript:

Over the time required to acquire the large number of single-molecule images needed for a super-resolution reconstruction, the sample can drift several hundred nm in all three dimensions. Such

drift would severely degrade the reconstruction and blur nanoscale features. To facilitate 3D drift correction, ...

4. *Also, did the authors consider highly inclined and laminated optical sheet microscopy (HILO) for the illumination? The method is readily installed for inverted microscopes, does not need any additionally optical parts, and gives superior SBR. If the method is not suitable, the authors should maybe discuss in the text.*

We agree that HILO would have improved the SBR by background reduction in comparison to wide-field epi-illumination, as demonstrated by Tokunaga *et al.* Nat. Methods 5 (2008), and by Konopka *et al.* Plant J. 53 (2008), using the very similar method variable-angle epifluorescence microscopy (VAEM). However, in contrast to HILO and VAEM, in TILT3D the sheet light intensity is independent of the axial position, and the thickness of the light sheet is independent of the angle of the light sheet. To clarify this to the reader, we have added the following sentences to the introduction:

However, early light sheet methods, e.g. selective plane illumination microscopy (SPIM) [Huisken, 2004], were designed for low-magnification imaging of large samples, such as embryos. These methods are incompatible with imaging close to a cover slip using a high numerical aperture (NA) imaging objective, which is a requirement for high contrast single-molecule imaging of sub-cellular structures. Two early methods producing a thin tilted beam are a highly inclined and laminated optical sheet (HILO, pseudo-TIR) [Tokunaga, 2008] and variable-angle epi-fluorescence microscopy (VAEM) [Konopka, 2008]. However, in these techniques, the intensity, position, and depth of the pumping light pattern are highly coupled, in contrast to the method presented here. More recently, ...

5. *For the lack of novelty, I cannot recommend the current manuscript for publication in Nature Communications.*

It is unfortunate that the novelty of this manuscript was not fully appreciated, and we would like to improve this. All the changes in the Introduction described above further clarify the advantages of TILT3D compared to previous designs. We also have described the many novel aspects in the manuscript in the original text, particularly in the section titled *TILT3D design and performance*, and we summarize them again here.

1. The design of the optical setup is novel
 - a. Introducing the light sheet at an angle to the emission path on its own is not novel and has been used before in earlier designs such as HILO. However, introducing the light sheet at an angle using a simple glass chamber and a glass prism is a novel design on its own.
 - b. This novel design allows the tilted light sheet to easily access the bottom surface of adherent cells. This eliminates the step that some research groups have pursued, i.e. suspension of the cells in special extracellular matrices or gels, or use of pairs of objectives at 45 degrees to the interface in order to image the entire cell.
 - c. Because of the simple design of the optical setup, we can, in principle, reposition the light sheet at any position (X, Y, or Z) within the sample chamber and still retain the same sectioning capability of the light sheet. In some earlier light sheet designs, such

- as HILO, the light sheet becomes thicker and much more tilted the farther away the biological sample is from the glass coverslip.
- d. The simplicity of this design allows future researchers to add this light sheet onto the standard inverted microscopes that they may already have. This is due to the collection objective still being located below the sample instead of being at an angle above or beside the sample, and also because there are few custom made components in the setup.
 - e. This design also greatly reduces the risk of contamination. Unlike previous light sheet designs, TILT3D does not require the immersion of a foreign object (such as another objective or an AFM cantilever) into the cell media or blinking buffer. This reduces the risk of introducing biological contaminants that will affect long-term live cell imaging or even fluorescent contaminants that will increase the background, a critical issue in sensitive single-molecule imaging.
2. Long axial range point spread functions (PSFs) are used
- a. The DH-PSF and Tetrapod PSFs are used because they have a longer axial range with high and optimal localization precision as compared to previous PSFs used for light sheet super-resolution imaging (e.g. standard and astigmatic PSFs).
 - b. Using the DH-PSF, we are able to image and localize all fluorophores illuminated within the tilted light sheet because the working axial range of this PSF is comparable to the thickness of the tilted light sheet. If the standard or astigmatic PSFs were used instead, the fluorophores that are too high or too low with respect to the nominal focal plane would result in PSFs that are too blurry to be localized.

By combining the novel tilted light sheet design with the long axial range PSFs, we have created a new capability of imaging and localizing single molecules in thick mammalian cells with an improved signal-to-background ratio and an increased working axial range.

We have made the following additional changes to the text to make the novelties of the TILT3D method more explicitly stated in the paper:

The long working distance allows the light sheet to be introduced without dipping the objective into the sample chamber, reducing the risk of **any biological or fluorescent** sample contamination.

A 10° downward tilt of the light sheet enables the light sheet to be introduced into the sample chamber far away from the distorting bottom glass-water interface, while allowing illumination even at the bottom **surfaces of adherent cells**.

6. *Minor points:*

1 In the section “3D imaging with long axial range DH-PSF”, it is not clear to me why the authors chose to exchange the DM by the DH phase mask. The section should be written more clearly.

We thank the reviewer for pointing out that this was unclear. The phase pattern of the DH-PSF has discontinuities (“phase jumps”) which are difficult to create using the continuous membrane of a deformable mirror. Therefore, we replaced the deformable mirror with a transmissive dielectric phase mask when imaging with the DH-PSF. Imaging with the DH-PSF can also be achieved using a phase-only liquid crystal spatial light modulator as was done in the first papers (Refs. 17-19). To clarify this we have added the following sentences to the *Optical setup* part of the *Methods* section:

Transmissive phase masks were used to implement the spatially non-smooth phase pattern of the DH-PSF. Since the DM consists of a continuous membrane, it is more suited to encoding smoothly varying phase patterns, such as those of the Tetrapod PSFs.

2 *Ref. 35 is not properly formatted*

We thank the reviewer for pointing this out.

We have corrected reference 35 (Halpern *et al.* 2015, now reference 38).

3 *A similar publication describes the use of light-sheet microscope and PSF engineering and should be considered for citation (Bin Yu, Yu, J., Li, W., Cao, B., Li, H., Chen, D., & Niu, H. (2016). Nanoscale three-dimensional single particle tracking by light-sheet-based double-helix point spread function microscopy. Applied Optics, 55(3), 449–453)*

We thank the reviewer for this suggestion. In the work by Yu *et al.* they track a single, bright 210 nm bead in an agarose solution using the DH-PSF. To improve contrast they illuminate the bead using conventional selective plane illumination microscopy (SPIM). This method is not compatible with imaging close to a coverslip using a high NA detection objective, and in this study they used a 20x 0.75 NA objective. This setup is thus not well suited for imaging of single molecules, and the authors only demonstrate tracking of a single, very bright emitter. This issue is resolved using TILT3D.

We have added a citation to this reference in the introduction about the DH-PSF.

Reviewer 3

We thank the reviewer for the time spent reading our paper and for their positive response to our manuscript.

Below, we provide responses (in red) to each of the questions/comments (*in black italic*), with our additions to the text indicated (in blue).

- The authors address an important problem in their manuscript the imaging of single molecules in three dimensions for the purpose of super resolution microscopy. In 3D samples background fluorescence can be a significant problem. Therefore selective illumination approaches have been proposed over recent years to reduce such out of focus background signals. Light sheet microscopy has received considerable attention with several different approaches having been published. The authors add a new approach which combines light sheet microscopy on the excitation side with the detection of engineered point spread functions.*

The paper is very carefully written and every step is well explained. Considering the number of alternative approaches using light sheet illumination that have already been published, I would, however, have expected a more detailed discussion of the pros and cons of this new approach to implementing light sheet microscopy.

We have extended the comparison between different light sheet designs in the introduction, as well as clarified some of the benefits of TILT3D in the *Results* section of the manuscript. Please see the responses to comments 4. and 5.

Reviewers' comments:

Reviewer #2 (Remarks to the Author):

The combination of a light sheet with glass chamber was already demonstrated by "Ritter, J. G., Veith, R., Veenendaal, A., Siebrasse, J. P., & Kubitscheck, U. (2010). Light Sheet Microscopy for Single Molecule Tracking in Living Tissue. PLoS ONE, 5(7), e11639–9".

The use of a prism was shown by "Greiss, F., Deligiannaki, M., Jung, C., Gaul, U., & Braun, D. (2016). Single-Molecule Imaging in Living Drosophila Embryos with Reflected Light-Sheet Microscopy. Biophysical Journal, 110(4), 939–946".

A tilted light sheet with glass chamber (though quite similar to HILO) was reported by "Hu, Y. S., Zhu, Q., Elkins, K., Tse, K., Li, Y., Fitzpatrick, J. A. J., et al. (2013). Light-sheet Bayesian microscopy enables deep-cell super-resolution imaging of heterochromatin in live human embryonic stem cells. Optical Nanoscopy, 2(1), 7".

I believe that the number of already published methods similar to this manuscript makes a thorough characterization/discussion of pros/cons of TILT3D necessary. For example, the authors claim that the movement of the tilted light sheet to any position within the sample chamber retains the same sectioning capabilities. I would like to see this statement as data. If I understand correctly, Fig. S2 hints towards this direction, but only shows data on the width as function of y . What about the thickness in x , y , and z ? I would expect some degree of distortion because of the tilted air-glass and glass-water interface, and the mm-sized chamber.

Reviewer 2

We thank the reviewer for providing detailed comments and suggestions for improving the manuscript. We have explicitly addressed each comment to better convey the pros and cons of TILT3D compared to previous techniques.

Below, we provide responses (in red) to each of the questions/comments (in *black italic*), with our additions to the text indicated (in blue).

1. *The combination of a light sheet with glass chamber was already demonstrated by “Ritter, J. G., Veith, R., Veenendaal, A., Siebrasse, J. P., & Kubitscheck, U. (2010). Light Sheet Microscopy for Single Molecule Tracking in Living Tissue. PLoS ONE, 5(7), e11639–9”.*

Disadvantage with this technique:

- They do not tilt their light sheet and can therefore not image all the way to the coverslip without distorting the light sheet at the interface at the bottom surface. Their implementation prevents imaging of most adherent mammalian cells, which are on the order of 10 μm thick.

This issue is solved using TILT3D. We have added a reference to this paper as shown in point 4 below.

2. *The use of a prism was shown by “Greiss, F., Deligiannaki, M., Jung, C., Gaul, U., & Braun, D. (2016). Single-Molecule Imaging in Living Drosophila Embryos with Reflected Light-Sheet Microscopy. Biophysical Journal, 110(4), 939–946”.*

Disadvantages with this technique:

- This method (and the original idea (Gebhardt *et al.* (2013) Nat. Methods 10(5) 421-426)) suffers from a ~ 2 μm inaccessible gap at the coverslip that cannot be illuminated, which prevents imaging of the lower parts of mammalian cells.
- The illumination objective is dipped into the sample chamber, which increases the risk of both biological and fluorophore contaminations.
- The cell must be (manually) positioned right next to a microprism before imaging.

All these disadvantages are avoided using TILT3D. We have added a reference to this paper as shown in point 4 below.

3. *A tilted light sheet with glass chamber (though quite similar to HILO) was reported by “Hu, Y. S., Zhu, Q., Elkins, K., Tse, K., Li, Y., Fitzpatrick, J. A. J., et al. (2013). Light-sheet Bayesian microscopy enables deep-cell super-resolution imaging of heterochromatin in live human embryonic stem cells. Optical Nanoscopy, 2(1), 7”.*

Disadvantages with this technique:

- Similar to HILO/VAEM, already discussed in last version of manuscript.
- The detection objective is dipped into the sample chamber, which increases the risk of sample contamination, as discussed in point 2.

These disadvantages are avoided using TILT3D. We have added a reference to this paper as shown in point 4 below.

4. *I believe that the number of already published methods similar to this manuscript makes a thorough characterization/discussion of pros/cons of TILT3D necessary.*

No previously published paper shows all the advantages that we provide with TILT3D. To further explicitly clarify the pros and cons of TILT3D compared to previous techniques, we have made the following additional changes to the manuscript:

We have added the reference to Ritter *et al.* in the introduction:

“These methods are incompatible with imaging close to a cover slip [Ritter *et al.* PLoS ONE, 5(7), e11639–9 (2010)] using a high numerical aperture (NA) imaging objective...”

We have more explicitly stated the benefits of TILT3D in the introduction, as a complement to the more detailed description in the *Results* section. We have also added references to the Greiss *et al.* and the Hu *et al.* studies:

“More recently, numerous light sheet designs have been implemented for SR imaging²⁸⁻³³, but these designs have drawbacks in certain situations. Some designs are incompatible with imaging of fluorophores close to the coverslip using high NA imaging objectives^{30, 31} [Greiss *et al.* Biophys. J. 110(4), 939–946 (2016)]. In some cases either the illumination or the detection objective is dipped into the sample chamber²⁹⁻³⁰ [Greiss *et al.* Biophys. J. 110(4), 939–946 (2016), Hu *et al.* Optical Nanoscopy, 2(1), 7 (2013)]. This increases the risk of both biological and fluorophore contaminations of the sample. Some previous designs require complicated optical and electronic apparatus or many custom-made parts which are often expensive and difficult to build and operate, and thus may not be easily accessible to the general research community.

Here, we present TILT3D, an imaging platform that combines a novel, *tilted* light sheet illumination strategy with long axial range PSFs. We alleviate many of the aforementioned difficulties in existing light sheet designs by tilting the illumination plane. The tilt allows for sectioning and imaging of cells all the way down to the coverslip. Two perpendicular objectives in close proximity are not required, which enables imaging using a high NA detection objective. No dipping of the objectives into the sample chamber is necessary, which reduces the risk of sample contamination. TILT3D: (a) yields high localization precision of single molecules in 3D over the entire axial range of a mammalian cell via a stack of light sheet slices combined with imaging with engineered PSFs in each slice, (b) has the usual light sheet advantages of reduced photobleaching and photodamage of the sample, but most importantly (c) is easy and cost-efficient to implement and operate.”

5. *For example, the authors claim that the movement of the tilted light sheet to any position within the sample chamber retains the same sectioning capabilities. I would like to see this statement as data. If I understand correctly, Fig. S2 hints towards this direction, but only shows data on the width as function of y. What about the thickness in x, y, and z? I would expect some degree of distortion because of the tilted air-glass and glass-water interface, and the mm-sized chamber.*

In Figure S2, we characterized the (axial) thickness in z of the light sheet as well as the in-plane width in y. Our light sheet enters the chamber in a very simple way through the parallel plate side of the chamber. For all axial (z) positions of the light sheet, the beam passes through the same optics and the same interfaces. Because the pivoting of the incoming beam occurs about a point in the back focal

plane, there is no angle change where the beam enters the prism, only a lateral translation, so the shape of the light sheet cannot change.

Proof that our light sheet does not change for different slices is in the 3D data of an extended cellular sample. To experimentally show this as data, we have added a new supplementary figure demonstrating that the fluorescence intensity and contrast at the rim of the nuclear lamina are independent of axial position of the light sheet. If the light sheet thickness was affected by axial translation, then both the brightness and contrast would have changed between the different sections.

Supplementary Figure 3 | Comparing different cell slices by axial translation of light sheet. The light sheet was moved together with the image plane using a motorized mirror, which allowed for sectioning throughout the entire thickness of a mammalian cell. Images show comparisons between light sheet (LS) and epi- (Epi) illumination at 2 μm and 4 μm from the coverslip. The graph shows line scans over the corresponding lines in the images, showing that the peak brightness and contrast are the same for different slices. All images show lamin B1 immunolabeled with Alexa Fluor 647 in a HeLa cell. All images are shown with the same linear grayscale. Scale bar is 5 μm .